# Interplay of structural preorganization and conformational sampling in UDP-glucuronic acid 4-epimerase catalysis

Christian Rapp[1], Annika Borg [1] & Bernd Nidetzky [1,2] ✉

Understanding enzyme catalysis as connected to protein motions is a major challenge. Here, based on temperature kinetic studies combined with isotope effect measurements, we obtain energetic description of C-H activation in NAD-dependent UDP-glucuronic acid C4 epimerase. Approach from the ensemble-averaged ground state (GS) to the transition state-like reactive conformation (TSRC) involves, alongside uptake of heat ($\Delta H^{\ddagger} = 54\,kJ\,mol^{-1}$), significant loss in entropy ($-T\Delta S^{\ddagger} = 20\,kJ\,mol^{-1}$; 298 K) and negative activation heat capacity ($\Delta C_p^{\ddagger} = -0.64\,kJ\,mol^{-1}\,K^{-1}$). Thermodynamic changes suggest the requirement for restricting configurational freedom at the GS to populate the TSRC. Enzyme variants affecting the electrostatic GS preorganization reveal active-site interactions important for precise TSRC sampling and H-transfer. Collectively, our study captures thermodynamic effects associated with TSRC sampling and establishes rigid positioning for C-H activation in an enzyme active site that requires conformational flexibility in fulfillment of its natural epimerase function.

Enzymes are exceptionally powerful catalysts of chemical transformations[1,2]. They achieve enormous rate accelerations[3] and do so with exquisite specificity[4]. Figuring out what makes enzymes so efficient is a subject of great fundamental interest[5–12] and of high practical importance[13–18]. Proposals on the origin of enzyme catalytic power agree on the requirement of a structurally preorganized polar active site for precise positioning of the reacting groups and for providing electrostatic stabilization[5]. Many of the current proposals also invoke dynamics (vibrational coupling) in enzyme function[6,19–23]. Global stochastic motions of protein structure are thought to converge at the active site to bring about optimal preorganization. Local fluctuations of the active site can enable the sampling of enzyme-substrate conformers that have electrostatics and internuclear distances precisely tuned for bond cleavage/formation[6,19,20,24]. However, the idea of enzyme catalysis based on protein vibrational modes coupling to the chemical coordinate is controversial[25–30].

A defining feature of enzyme catalysis is mechanistic cooperativity among the individual active-site components in lowering the activation free energy for the reaction[31]. Comprehensive description of the

catalytic effect in the enzyme is challenging, not only in terms of the cooperative energetics but also regarding the possible role of dynamics. Biophysical studies, by hydrogen-deuterium exchange[32,33] and room temperature x-ray crystallography[34–38] in particular, support a conformational ensemble view of the enzyme function. The free-energy conformational landscape involves multiple interchanging protein states that can differ in activity[34]. Adiabatic and non-adiabatic descriptions of quantum mechanical hydrogen tunneling associated with enzymatic C-H activation reactions provide a direct physical probe of the experiment into the connection between local fluctuations of the preorganized active site, affecting the sampling of donor-acceptor distances (DAD) suitable for the H-transfer, and catalysis of the chemical step[6,19,22,39]. The view that enzyme catalysis involves stochastic sampling of reactive protein conformational substates is expressed in Eq. (1).

$$k_{obs} = f_R k_R \tag{1}$$

$k_{obs}$ is the ensemble-averaged experimental rate, $f_R$ is the fraction of total enzyme occupying the set of active enzyme-substrate

[1]Institute of Biotechnology and Biochemical Engineering, Graz University of Technology, NAWI Graz, 8010 Graz, Austria. [2]Austrian Centre of Industrial Biotechnology (acib), 8010 Graz, Austria. ✉e-mail: bernd.nidetzky@tugraz.at

substates, and $k_R$ is the corresponding rate constant for the H-transfer[6,34]. Based on seminal Jencksian concepts on enzyme catalysis[40] (see also discussion on entropy of Åqvist and co-workers[11]), Klinman and co-workers[2,40,41] additionally put forward the idea of entropic penalty associated with restricting the degrees of conformational freedom to those sub-states of enzyme that can give rise to catalysis (the $f_R$ population). Extensive studies of alcohol dehydrogenases show that for oxidation of benzyl alcohol by NAD$^+$ in the optimum temperature range of the enzyme used, the activation entropy is positive significantly ($-T\Delta S^\ddagger$ of up to 30 kJ/mol)[19,42–44]. The positive $-T\Delta S^\ddagger$ was interpreted to indicate a relatively greater degree of disorder (flexibility) at the ensemble-averaged ground state (GS) compared to the H-transfer crossing point, consistent with an entropy funnel model of conformational sampling to populate the transition state-like reactive conformation (TSRC)[41,45,46]. This model is strongly opposed by Warshel and co-workers[25–28] who explain the $-T\Delta S^\ddagger$, and the change thereof when moving outside of the enzyme's optimum temperature range, by a reorganization of protein polar groups in response to change in the charge distribution of the reacting system going from the GS to the transition state[47]. Considering the controversial views, we figured that fresh evidence from the study of a different but relevantly similar (alcohol dehydrogenase-like) enzyme would be important to gain an advanced perspective. UDP-glucuronic acid 4-epimerase (UGAepi)[48–51] seemed to be an excellent candidate to analyze the functional interconnection between active-site preorganization, conformational sampling and energetics of the catalytic reaction.

UGAepi catalyzes C4 epimerization of UDP-glucuronic acid (UDP-GlcA), yielding UDP-galacturonic acid (UDP-GalA) for cell wall polysaccharide biosynthesis in plants and microbes[48,50]. Mechanistically, the reaction proceeds by oxidation-reduction at the C4, via hydride transfer to and from tightly bound nicotinamide coenzyme (Fig. 1a). Each hydride transfer is facilitated by a concerted proton transfer[48,50,51]. The transiently formed UDP-4-keto-hexuronic acid is rotated in the active site so that keto-group reduction can happen from both faces of the carbonyl. In order to generate the stereo-inverted product, therefore, the chemical coordinate in the reacting enzyme must include conformational sampling on an (orthogonal) protein configuration coordinate. Considering the requirement of conformational flexibility for UGAepi function, the enzyme active site is structurally preorganized in remarkable degree (Fig. 1b). It achieves precise positioning of both sugar C4 epimers without the need for side chain rearrangements (Supplementary Fig. 1)[50]. Repositioning of the 4-keto-hexuronic acid moiety involves extensive sampling of sugar ring pucker, yet the active-site configuration is almost unchanged in the process[51]. Biochemical study of the *Bacillus cereus* UGAepi establishes that C-H activation for substrate oxidation is rate-limiting for the overall C4 epimerization of UDP-GlcA[48].

The essential feature that the experimental $k_{cat}$ reflects a single chemical step of the enzymatic mechanism has enabled us here to apply temperature dependence studies and isotope effect measurements to probe the energetics and conformational characteristics of UGAepi activity[51]. Based on thermodynamic activation parameters, we find that approach from the ensemble-averaged GS to the TSRC involves restricting the configurational freedom of enzyme-substrate complex in considerable degree ($-T\Delta S^\ddagger \leq 44$ kJ/mol at 25 °C). Using mutagenesis, we succeed in dissecting the roles of local electrostatic interactions at the GS and TSRC sampling. Importantly, we find that perturbation of the chemical coordinate of proton transfer (Fig. 1a), caused by site-specific substitution of the catalytic general base (Y149), hugely impacts the hydride transfer coordinate by energetic destabilization while it affects only weakly the sampling at the TSRC on the protein coordinate. Conversely, we identify mutation (S127A) that creates a large specific perturbation of TSRC sampling due to conformational flexibility with only marginal effect on the free-energy

barrier of the reaction. Overall, therefore, UGAepi promotes H-transfer from UDP-GlcA by combining sampling with electrostatic stabilization for precise positioning at a conformationally rigid TSRC. Interplay between active-site preorganization and conformational sampling is thus required not only in the overall UGAepi reaction but also in the immediate chemical step.

## Results

### Temperature dependence of KIEs connects active-site preorganization to TSRC sampling mode

Kinetic isotope effects (KIEs) are powerful mechanistic probes of H-transfer reactions. The temperature dependence of KIEs was proposed to provide signature information on the DAD sampling at the tunneling ready state (TRS) that controls the quantum mechanical tunneling associated with the H-transfer[6,19,52,53]. In the interpretation of the authors, the idea of TRS is analogous to that of TSRC[54], involving sampling of specific conformations for the chemical step to occur. Conformational sampling in general affects both $f_R$ and $k_R$ in Eq. (1), but as isotopically insensitive terms are effectively canceled out in the KIE, the results are more straightforward to interpret than the $k_{obs}$[6].

Here, we determine primary deuterium KIEs from direct comparison of $k_{cat}$ for UDP-GlcA substrate having [$^1$H] or [$^2$H] at the reactive C4. KIEs are recorded in the temperature range 4–50 °C. The native UGAepi is compared to a set of site-directed variants. Each variant involves a local perturbation of electrostatic preorganization at the GS (Fig. 1b) that is expected to result in the impairment of a distinct mechanistic feature of the catalysis, as follows. Y149 is directly involved in the chemical step (Fig. 1a). The Y149F variant retains a low level of epimerase activity (~0.01% of wild-type) despite acid-base catalytic group removed. Its activity is explained by a water molecule, bound at the position of the Y149 phenolic hydroxyl, enabling rescue of the catalytic function[50]. R88 and P85 belong to flexible elements of UGAepi structure[50]. R88 is part of a protein conformational change (loop Pro85-Trp91) coupled to substrate binding and required for GS preorganization. As shown in Fig. 1b, the R88 side chain is tightly bonded to the α-phosphate group of UDP-GlcA. P85 is involved in a peptide bond (P85-G86) flip that represents the only change in active-site conformation associated with the product GS compared to the substrate GS (Fig. 1b, Supplementary Fig. 1)[50]. S127 represents rigid positioning of the substrate for control of reaction selectivity. It contributes to holding the C5 carboxylate in an equatorial orientation (Fig. 1b) which disfavors decarboxylation of the chemically labile 4-keto-intermediate on stereo-electronic grounds[51].

We show for each variant, and confirm for native UGAepi, that during reaction with UDP-GlcA at steady state, the portion of total enzyme present in the reduced NADH form is not increased compared to the as-isolated enzyme in the resting state (≤3.5%; Supplementary Table 1). This evidence, together with the observation of a substantially normal KIE on the $k_{cat}$ (Table 1), identifies C-H activation for substrate oxidation as the common rate-determining step in all enzymes.

Results in Fig. 2 reveal subdivision of the enzymes according to temperature-independent (wild-type, Y149F; panel a) and strongly temperature-dependent KIEs (P85G, S127A; panel b). The KIE of R88A is weakly temperature-dependent, as shown in Fig. 2a. Interestingly, the magnitude and temperature characteristics of KIE appear to be unrelated to the degree of disruptive effect of mutation on the $k_{cat}$ (Table 1).

The Y149F variant is lowest in $k_{cat}$ ($9.2 \times 10^3$-fold decrease) while it retains the wild-type property of temperature-independent KIE. The S127A variant has the strongest temperature dependence of KIE among the enzymes used, but its $k_{cat}$ is hardly affected (2.3-fold decrease) compared to native UGAepi. The KIE of S127A at 25 °C ($^D k_{cat} = 3.52$) exceeds considerably that of Y149F ($^D k_{cat} = 2.67$). The

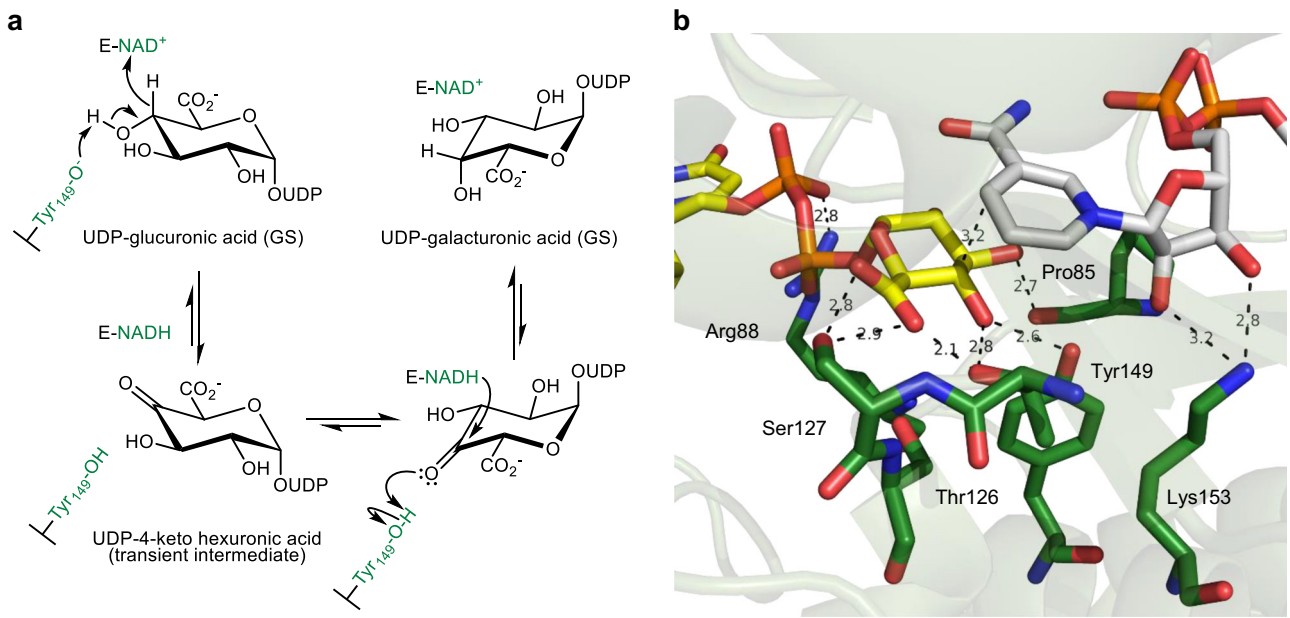

**Fig. 1 | Enzymatic C4 epimerization of UDP-GlcA by UGAepi. a** Proposed reaction mechanism and (**b**) preorganized active site of the enzyme observed crystallographically. Carbon atom coloring: enzyme (green), UDP-GlcA (yellow), NAD⁺ (gray). GS ground state.

**Table 1 | Parameters of C-H activation in UDP-GlcA by UGAepi and site-directed variants thereof**

| Parameter | Wild-type | R88A | Y149F | S127A | P85G |
|---|---|---|---|---|---|
| $k_{cat}$ $(s^{-1})^a$ | 5.60 (±0.06) ×10⁻¹ | 2.62 (±0.02) ×10⁻² | 6.10 (±0.05) ×10⁻⁵ | 2.42 (±0.01) ×10⁻¹ | 1.65 (±0.03) ×10⁻² |
| KIE $(^{D}k_{cat})^a$ | 1.97 ±0.26 | 3.83 ±0.10 | 2.67 ±0.32 | 3.52 ±0.20 | 3.31 ±0.32 |
| $-T\Delta S^{‡}$ $(kJ\,mol^{-1})^{a,b}$ | 20 ± 3 (23 ± 3)ᶜ | 44 ± 4 (45 ± 4)ᶜ | 41 ± 4 (46 ± 4)ᶜ | 42 ± 4 (42 ± 4)ᶜ | 43 ± 6ᵈ (56 ± 3)ᶜ, ᵈ |
| $\Delta H^{‡}$ $(kJ\,mol^{-1})^b$ | 54 ± 3 (53 ± 3)ᶜ | 38 ± 4 (40 ± 4)ᶜ | 55 ± 4 (53 ± 4)ᶜ | 34 ± 4 (38 ± 3)ᶜ | 40 ± 6ᶜ, ᵈ (30 ± 4)ᶜ, ᵈ |
| $\Delta C_p^{‡}$ $(kJ\,mol^{-1}\,K^{-1})^b$ | −0.64 ± 0.06 (−0.50 ± 0.06)ᶜ | −0.20 ± 0.06 (0.07 ± 0.02)ᶜ | −1.11 ± 0.12 (−1.16 ± 0.13)ᶜ | −3.62 ± 0.54 (−1.99 ± 0.16)ᶜ | n.d. (n.d.) |

ᵃDetermined at 25 °C.
ᵇFrom fits of Eq. (2) to temperature profiles of $k_{cat}$.
ᶜData for the [²H] substrate.
ᵈFrom Eyring fits (ΔC_p‡ = 0) of the data in the range 4–35 °C.

KIE-temperature profiles of P85G and S127A (Fig. 2b) show maximum at intermediate temperature and decrease above and below that temperature. KIE decrease upon increase in $k_{cat}$ at high temperature is a known phenomenon in enzymes, those harboring mutations in particular[6,19,55]. It is explainable by nonoptimal DAD sampling modes, involving DAD population(s) centered at an elongated H-tunneling distance less suitable for the heavy isotope[6,39]. However, work by Warshel and co-workers[56] has reproduced the experimental temperature dependence of KIE in wild-type and variant forms of dihydrofolate reductase (i.e., increase of KIE with decrease in temperature) and describe it in terms of the temperature dependence of the dominant DAD. Decrease at low temperature, as observed for S127A and P85G (Fig. 2b), represents a largely unexplored feature of KIE in enzymatic H-transfer[57,58] and cannot be accounted for by full-tunneling models[6,39]. It potentially reflects shortening of the dominant H-transfer distance upon protein conformational rigidification at low temperature. Note: A DAD of 2.77 Å was calculated for the transition state in a classical (over-the-barrier) description of the enzymatic H-transfer by UGAepi[51]. Full-scale computational study by molecular simulation will probably be necessary to unravel the origin of KIE decrease at low temperature.

## Solvent isotope effects suggest substrate oxidation through a concerted reaction

Evidence from QM/MM metadynamics calculations supports a concerted mechanism of substrate oxidation by wild-type UGAepi (Fig. 1)[51]. We considered that due to removal of the catalytic base residue of the enzyme, the Y149F variant might involve change in the mechanism so that the proton transfer becomes more strongly uncoupled from the hydride transfer. To address this question, we measured solvent kinetic isotope effects (SKIE) on the $k_{cat}$ for reaction of wild-type and Y149F with both ¹H and ²H isotope substrates. Multiple isotope effects due to deuteration of substrate and solvent are useful to distinguish between concerted and stepwise proton and hydride transfer in dehydrogenase reactions[59–61]. The wild-type exhibits a $^{D2O}k_{cat}$ of 1.27 ± 0.10 and 1.25 ± 0.07 for the reaction with the ¹H and ²H substrate, respectively. A $^{D2O}k_{cat}$ of 2.02 ± 0.18 (¹H substrate) and 2.02 ± 0.12 for (²H substrate) was determined for Y149F. The substrate KIE values ($^{D}k_{cat}$) are identical with limits of error in ²H₂O and ¹H₂O buffer (wild-type: 2.0; Y149F: 2.7) and are larger in magnitude than the corresponding solvent KIEs ($^{D2O}k_{cat}$). Normal SKIEs ($^{D2O}k_{cat} > 1$) imply that one or more steps of the reaction of wild-type and Y149F are sensitive to solvent deuteration. The information contained in the observed SKIE

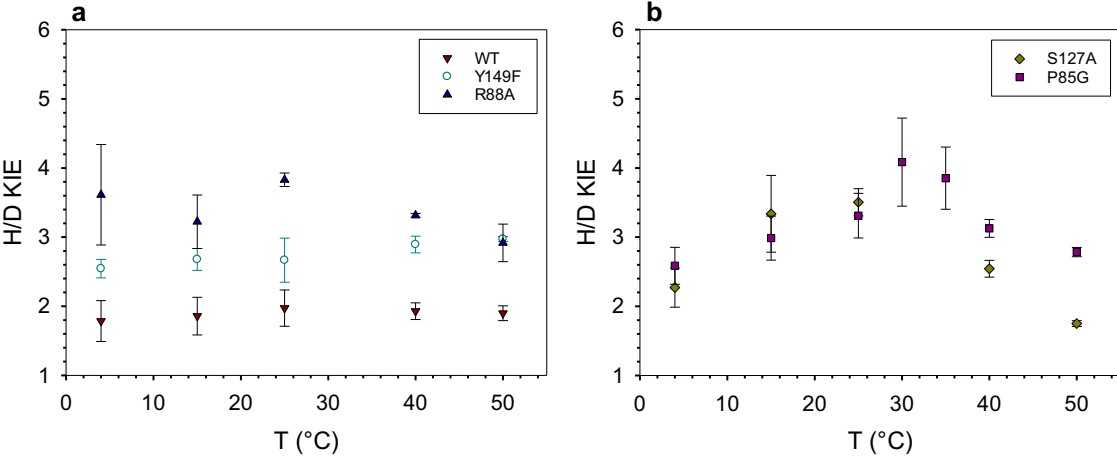

**Fig. 2 | Temperature profiles of KIE on $k_{cat}$ for UGAepi and variants thereof.** Panels group enzymes showing weak (**a**) and strong (**b**) temperature dependence of KIE. Symbols show data presented as mean values ± SD, with $n = 3$ biologically independent samples examined over 3 independent experiments. WT wild-type.

may be complex and not disclose the presence or absence of rate limitation by the proton-transfer step. However, it is significant that substrate KIEs are unaffected by the change of solvent $^1H_2O$ to $^2H_2O$. The result implies that in both wild-type and Y149F, the catalytic step of hydride transfer is independent of the step of the enzymatic mechanism that exhibits the solvent sensitivity. It suggests furthermore that both isotope effects ($^Dk_{cat}^{D2O}$, $k_{cat}$) arose from a single rate-limiting step, consistent with a concerted mechanism of the wild-type enzyme and retention of the same in the Y149F variant.

### Reaction energetics revealed in temperature-rate profiles

Temperature dependencies of $\ln k_{cat}$ exhibit downward curvature at high temperature in a degree that varies with enzyme and substrate isotope used (Fig. 3). Common reasons for deviation from classical Arrhenius-like (linear) behavior are ruled out here[18,30,62]. All UGAepi enzymes are completely resistant to irreversible inactivation up to 50 °C. The wild-type UGAepi exhibits a melting temperature ($T_m$) of 57 ± 2 °C, consistent with the $T_m$ of 62.4 °C calculated from the enzyme structure (Supplementary Fig. 10). The UGAepi variants are predicted to have slightly enhanced $T_m$ of 66–67 °C or higher (Supplementary Table 3).

Loss of activity due to reversible change in protein structure at high temperature cannot reasonably be dependent on substrate isotope. No enzyme-NADH accumulates at high temperature, consistent with substrate oxidation as the rate-limiting step in the full temperature range used. Importantly, similar results were obtained at low temperature (4 °C) for reactions of P85G and S127A. The two enzymes required deeper interrogation due to the conspicuous temperature profiles of their $k_{cat}$ (Fig. 3c, d) and $^Dk_{cat}$ (Fig. 2b). The catalytic C-H activation involves proton transfer concerted with the hydride transfer (Fig. 1a[51]; see the SKIE results above), rendering the idea of kinetic separation of the two chemical steps pointless. Moreover, a precatalytic conformational step in the mechanism might be temperature dependent, yet it would not be sensitive to deuteration of the substrate. Therefore, these considerations strongly support the suggestion that the observed temperature profile of $k_{cat}$ reflects change of the free energy barrier of the catalytic step in response to increase of temperature.

Development of curvature in the profile might thus be attributed to an apparent activation heat capacity ($\Delta C_p^{\ddagger}$; Eq. (2)) for the chemical step[24,63–65].

$$\ln k_{cat} = \ln \frac{k_B T}{h} - \frac{[\Delta H_{T0}^{\ddagger} + \Delta C_p^{\ddagger}(T - T_0)]}{RT} + \frac{[\Delta S_{T0}^{\ddagger} + \Delta C_p^{\ddagger} \ln(T/T_0)]}{R} \quad (2)$$

$R$ is gas constant (kJ mol$^{-1}$ K$^{-1}$), $\Delta H^{\ddagger}$ is enthalpy of activation (kJ mol$^{-1}$), $\Delta S^{\ddagger}$ is entropy of activation (kJ mol$^{-1}$ K$^{-1}$), $k_B$ is Boltzmann constant (1.381 × 10$^{-23}$ J K$^{-1}$), $h$ is Planck constant (6.626 × 10$^{-34}$ Js), $T$ is temperature (K), $T_0$ is an arbitrary reference temperature (K), and $\Delta C_p^{\ddagger}$ is activation heat capacity (kJ mol$^{-1}$ K$^{-1}$). The idea of the model (Eq. (2)), termed macromolecular rate theory (MMRT)[24,63–66], is that of difference in energy fluctuations at the GS and TSRC. In the common situation that GS fluctuations are reduced at the TSRC, a negative $\Delta C_p^{\ddagger}$ emerges. MMRT has been used to analyze different enzymes, wild-type and mutant, with experimental and computational evidence shown in its support[24,65,67].

Before analyzing the current results (Fig. 3) with Eq. (2), however, we consider important objection of Åqvist's concerning the fundamental difficulty to discriminate MMRT from alternative kinetic models, in particular models that involve conformational sampling at the GS and explain decrease of $k_{cat}$ at high temperature due to temperature effect on lowering the fraction of reactive GS conformers ($f_R$; Eq. (1))[67–69]. Here, the temperature dependence of the KIE enables clear-cut model discrimination. While conformational sampling may give rise to differential reactivity of the substrate $^1H$ and $^2H$ isotope by way of effect on DAD fluctuations, its increased importance at elevated temperature (as implied by the kinetic model) is expected to result in a temperature-dependent increase in the KIE, contrary to the observation of KIE unchanged or decreased in the high temperature range. Moreover, we are not aware of publications showing that the decreased frequency of substrate vibrational modes due to substitution with heavy isotopes would translate into the protein energy landscape, such that conformational sampling between active ES and inactive ES' sub-states would be altered radically.

Table 1 summarizes the results of nonlinear fits of Eq. (2) to individual temperature profiles in Fig. 3. The temperature dependence of $k_{cat}$ for P85G involves a breakpoint between 35 °C and 40 °C (Fig. 3d). The low temperature range (≤35 °C) is fitted with a simple Eyring model ($\Delta C_p^{\ddagger} = 0$). The molecular origin of discontinuity in the $k_{cat}$ temperature profile of P85G was not pursued. It is not due to change of rate-limiting step, evidenced by absence of enzyme-NADH at steady state. Studies of other enzymes have explained similar breakpoints in temperature-rate profiles to arise from a temperature-dependent change in the equilibrium population of protein conformational sub-states (effectively $f_R$, see Eq. (1))[30,57,70,71]. In this context, we note a recent study[54] of Mulholland, Arcus and co-workers who expand the earlier MMRT approach to describe the temperature dependence of $\Delta H^{\ddagger}$ and $\Delta S^{\ddagger}$ (see Supplementary Fig. 2 for data from the current study) in terms of a kinetic model for the enzymatic reaction that includes conformational sampling as a precatalytic microscopic step

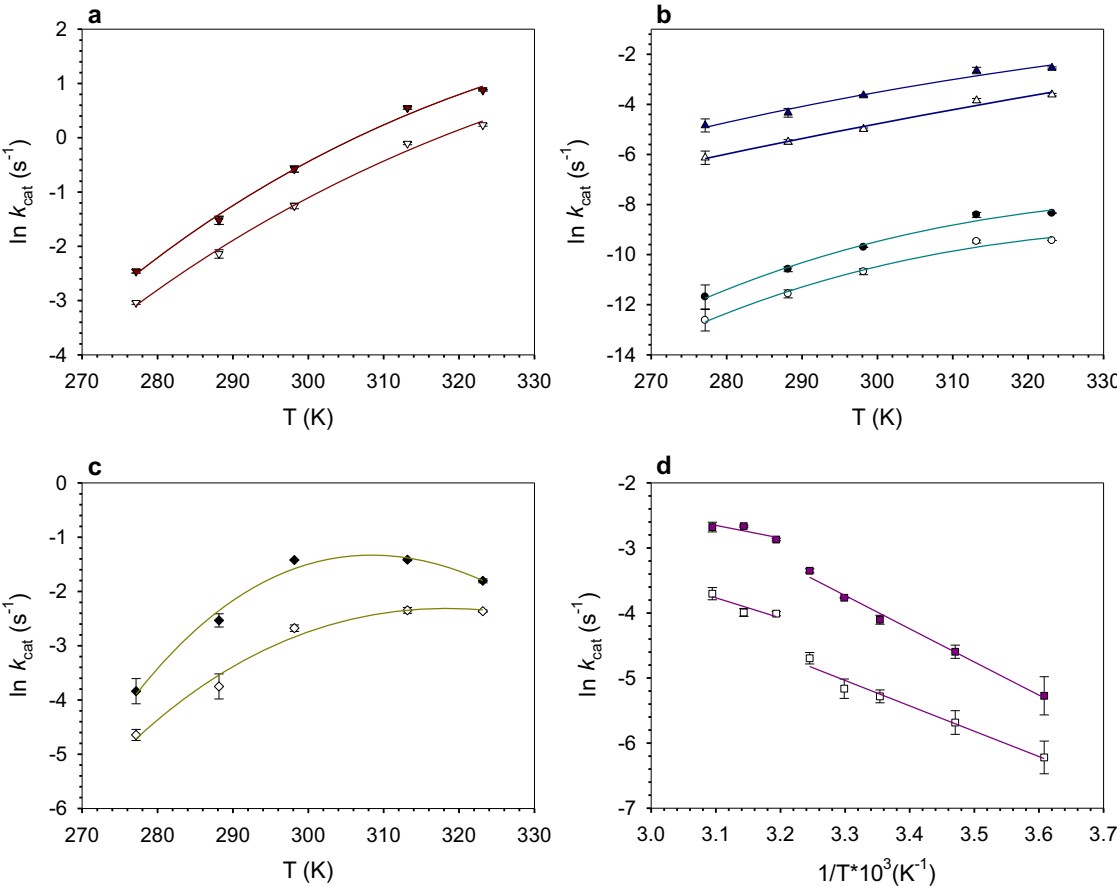

**Fig. 3 | Fits of temperature-rate profiles for UGAepi and variants thereof.** Curvature arises from fitting data of (**a**) wild-type, (**b**) R88A (blue), Y149F (turquoise) and (**c**) S127A to MMRT (Eq. (2)). Due to apparent breakpoint in the temperature profile, straight-line fits were used for P85G in the temperature range indicated in panel (**d**). Closed and open symbols represent [¹H] and [²H] isotopes, respectively. Data are presented as mean values ± SD, with $n = 3$ biologically independent samples examined over 3 independent experiments.

(GS → TSRC → transition state). The authors[54] discuss the role of conformational sampling in different scenarios of kinetic significance of this step. We have been careful to show with experiment that the assumption of multistep kinetic complexity on the measured $k_{cat}$ of UGAepi is not necessary (see above). The existence of additional kinetic steps not accessible to the experiment cannot be excluded though. In the proposed catalytic reaction of UGAepi, the TSRC undergoes turnover into product without the requirement for significant further uptake of energy. This view of TSRC may differ from that of Walker et al.[54] who appear to consider a sizable energy barrier between TSRC and the actual transition state. These points notwithstanding, we made an effort to also apply the expanded MMRT (Supplementary Eq. (2)) model to describe the temperature profiles of $k_{cat}$ for selected UGAepi enzymes (Supplementary Fig. 3). However, we realized that Supplementary Eq. (2) contains too many fit parameters for use with the available data sets (Supplementary Fig. 3). In summary, we suggest that the standard MMRT model (Eq. (2)) is well suitable to describe the temperature profiles in Fig. 3.

In the native UGAepi, approach from the GS to the TSRC involves substantial loss of entropy ($-T\Delta S^{\ddagger} = 20$ kJ mol⁻¹; 298 K), accounting for ~30% of the total free energy barrier for the process. In the reaction of ¹H and ²H substrate, the heat taken up is the same and entropy loss does not differ significantly. The $\Delta C_p^{\ddagger}$ is negative, slightly less so for the ²H substrate. Enzyme variants show a distinct pattern of change in the thermodynamic parameters relative to the native UGAepi and to each other. Y149F exhibits the same $\Delta H^{\ddagger}$ as the wild-type, yet the $-T\Delta S$ is increased strongly and the $\Delta C_p^{\ddagger}$ is more negative. The other variants involve enthalpy-entropy compensation ($\Delta H^{\ddagger}$ lowered, $-T\Delta S^{\ddagger}$ increased) compared to wild-type. Distinctive change embodied by S127A is on $\Delta C_p^{\ddagger}$ that is substantially more negative than in native UGAepi. R88A involves a less negative $\Delta C_p^{\ddagger}$ than wild-type. Common trend in all enzymes is that the $\Delta C_p^{\ddagger}$ is less negative for ²H compared to ¹H substrate. The entropic contribution of ²H compared to ¹H is factually indistinguishable among the enzymes.

## Discussion

### Conformational sampling and entropy change associated with UGAepi activity

The overall activation free energy ($\Delta G^{\ddagger}$) for substrate C-H activation by native UGAepi (at the temperature of enzyme characterization, 298 K) involves a considerable penalty ($-T\Delta S^{\ddagger}$) due to a sizable negative activation entropy. Supplementary Fig. 2 shows this to be true for a representative temperature range (293–303 K) despite the fact that both $\Delta H^{\ddagger}$ and $\Delta S^{\ddagger}$ are temperature dependent. Assessment of the results at a single temperature (298 K) is thus not only meaningful but also intuitively accessible. Irrespective of the magnitude of $\Delta G^{\ddagger}$ change relative to the wild-type reaction, the reactions of UGAepi variants uniformly embody altered enthalpy-entropy partitioning due to increased penalty ($-T\Delta\Delta S^{\ddagger} \geq 21$ kJ/mol; 298 K) from entropic effect (Table 1; see Supplementary Fig. 2 for data of S127A). Evidence from this study and earlier computational work by molecular simulations[51] suggests that conformational restriction towards the TSRC from a relatively more flexible GS might represent the mechanistic origin of the entropy penalty. The free-energy landscape (FEL) of the reacting

UGAepi obtained by QM/MM calculations reveals an ensemble of iso-energetic GS conformers that feature substantial fluctuation (sampling) of donor-acceptor atom distances of proton and hydride transfer[51]. Since the calculations assumed classical (over-the-barrier) mode of H-transfer, the barrier crossing point is effectively a transition state, revealed as a narrow saddle point region in the FEL[51]. QM/MM calculations are further informative in pointing to the requirement for restricting the substantial configurational freedom available at the GS in order to populate the TSRC for barrier crossing, consistent with the entropy penalty found experimentally. They hold additional insight in suggesting tight and rigid positioning at the TSRC, consistent with the implication of temperature-independent KIE that local distance sampling is relatively unimportant in the native UGAepi[6,19,39]. The negative $\Delta C_p^\ddagger$ derived from fit of the temperature-rate profile is furthermore consistent with the idea that energy fluctuations become reduced upon the approach from relatively flexible GS to rigid TSRC.

Further support to the idea of conformational sampling required in order to populate the TSRC comes from inspecting different enzyme subunits in the asymmetric unit of UGAepi crystal structures with bound UDP-GlcA or UDP-GalA. In these GS structures[50], the DAD varies between 2.8 and 3.3 Å (PDB: 6ZLL, 6ZLD). The overall variation in crystallographic DAD (|0.5 Å|) in the wild-type enzyme approaches the DAD (2.77 Å) obtained for the transition state and thus strongly supports fluctuating conformations[51]. Moreover, since the catalytic oxidation does not require sampling of any other than the $^4C_1$ chair conformation of the UDP-GlcA sugar moiety, the entropic contribution must largely arise from enzymatic configurational changes or sampling of catalytically productive protein sub-states. It is worth noting here the effect of $\Delta C_p^\ddagger$ on change of enthalpy-entropy partitioning upon increase in temperature. The apparent $-T\Delta S^\ddagger$ increases while the apparent $\Delta H^\ddagger$ decreases (Supplementary Fig. 2). This trend of change in activation parameters appears consistent with the proposed sampling of conformations restricting or releasing TSRC configurations, whereby elevated temperature releases GS motions more strongly than it can release restricted TSRC motions. The same overall trend of temperature dependence of $-T\Delta S^\ddagger$ and $\Delta H^\ddagger$ is predicted from molecular simulations of C-H activation by alcohol dehydrogenase[25,47,72], with the major difference however that the activation entropy $\Delta S^\ddagger$ is found to be significantly positive at low temperature and effectively zero in the optimum temperature range. The positive $\Delta S^\ddagger$ is explained by decrease in restriction on motions of active-site dipoles when the reacting enzyme proceeds from the polar GS to the less polar TS (transition state), as shown in Supplementary Fig. 4b[47]. Alcohol dehydrogenase differs from UGAepi in its use of $Zn^{2+}$ instead of a general catalytic base to facilitate deprotonation of the substrate alcohol group for C-H activation. Despite these differences, the trend change in charge distribution during transition from the GS to the TSRC is expected to be similar for the two enzymes (Supplementary Fig. 4). The negative activation entropy in UGAepi reactions cannot therefore be ascribed to this change and its origin in conformational sampling is supported.

### Effect of perturbation of electrostatic preorganization of GS

UGAepi variants are subdivided according to temperature dependence of KIE. Within the respective groups, Y149F and S127A stand out for the highly distinctive feature of mechanistic relevance to hydride transfer catalysis that each of the two variants embodies. Y149F represents defect in TSRC stabilization specifically whereas S127A is equally specific in representing impairment of conformational sampling at an energetically unaffected TSRC.

The increased free energy barrier for Y149F compared to native enzyme is exclusively due to enlarged penalty from a more strongly negative activation entropy. We interpret this change, together with the substantially more negative $\Delta C_p^\ddagger$ and the elevated $^{D2O}k_{cat}$ in the Y149F reaction, to arise from the added requirement in the enzyme variant to position water for proton transfer at the configurationally highly rigid TSRC. Structural placement of water was shown for the Y149F crystal structure in complex with UDP-4-deoxy-4-fluoro GlcA (Supplementary Fig. 5a; PDB: 6ZLJ)[50]. In agreement with this view, the crystal structure further suggests almost no change in GS pre-organization caused by the site-directed substitution (Supplementary Fig. 5)[50].

Analyzed at 298 K (see Supplementary Fig. 2), S127A involves large change in enthalpy-entropy partitioning within the frame of almost constant $\Delta G^\ddagger$ compared to native UGAepi. It features a substantially increased $-T\Delta S^\ddagger$ that is comparable in magnitude to that of Y149F. The entropic effect embodied by S127A is plausibly ascribable to a more flexible GS, and the added requirement of restricting configurational freedom upon transition to the TSRC, as compared to the wild-type enzyme. The $\Delta C_p^\ddagger$ substantially more strongly negative for S127A than wild-type enzyme is also consistent with the interpretation. The TSRC of S127A appears to be more loosely organized than that of native UGAepi and might comprise extensive conformational sampling. The particular temperature dependence of KIE in S127A, with decrease of KIE observed at both high and low temperature, might reflect the temperature dependence of the DAD, different for the $^1$H and $^2$H isotope. Thermally enhanced DAD sampling at the TSRC is assumed to cause decrease in KIE, as shown for numerous enzymes[6,55]. KIE decrease at low temperature probably arises from a shortening of the dominant DAD which in S127A can be enabled by a lowered $-T\Delta S^\ddagger$ penalty due to $\Delta S^\ddagger$ becoming less negative upon decrease in temperature (Supplementary Fig. 2c, d). This shortening of H-transfer distance would specifically benefit the transfer of the $^2$H isotope at low temperature relative to the temperature of maximum KIE. The $\Delta C_p^\ddagger$ of S127A is considerably more negative for the $^1$H isotope, consistent with the idea of shortened DAD due to TSRC fluctuations that are more strongly reduced at low temperature in the reaction of $^1$H compared to $^2$H substrate. Similarly, increased isotopic substitution of glucose reacted with glucose dehydrogenase was shown to induce less curvature in Arrhenius plots[58]. Effects were attributed to shifts in the conformational landscape leading to different GS-TSRC fluctuations[58]. Despite its $k_{cat}$ much higher than that of Y149F, S127A exhibits the more sizeable KIE of the two enzyme variants. Differences in the dominant DAD for the corresponding DAD distributions might explain the effect.

P85G and R88A involve consequences in activity overall comparable, but not as distinctively clear as, respectively, S127A and Y149F. Both P85G and R88A feature destabilization of the TSRC compared to wild-type ($\Delta\Delta G^\ddagger = 8-9$ kJ mol$^{-1}$), essentially due to entropic effect. P85G resembles S127A in its evident requirement of extensive conformational sampling at the TSRC. R88A also exhibits impaired positioning compared to wild-type enzyme. However, in contrast to S127A it appears to lack the flexibility required for TSRC sampling. The point is emphasized by the substantial $\Delta C_p^\ddagger$ of 3.4 kJ mol$^{-1}$ K$^{-1}$ between R88A and S127A. With its low $\Delta C_p^\ddagger$ and weak KIE temperature dependence, R88A rather resembles the wild-type and Y149F that both were shown to plausibly involve rigid positioning at the TSRC. R88A appears to embody defect in active-site preorganization, probably resulting from coupled-motion loop conformational change[50] impaired by the site-directed substitution. Since positioning in R88A seems to be still rather rigid as compared to S127A and P85G, this variant cannot rely on TSRC sampling in order to promote H-transfer from lower-energy protein sub-states.

### Implications for proposals of enzyme catalysis involving hydride transfer

C-H activation by UGAepi involves mechanistic features consistent with both the electrostatic preorganization and the conformational sampling view of enzyme catalysis. The primary source of UGAepi catalysis to the chemical step of hydride transfer appears to be

electrostatic in origin[5] and can likely be attributed to proton abstraction from the 4-OH by the general base of the enzyme (Y149). The GS places the UDP-GlcA substrate in a well-organized polar environment of active site. The donor-acceptor atom distances for hydride and proton transfer fluctuate by ~0.5 Å at the GS. Approach from the GS to the TSRC involves thermal activation as expected, but it also necessitates restriction in configurational freedom to a substantial degree[41,46]. The requirement for precise positioning of the donor-acceptor atoms for the hydride and proton transfer likely accounts for the entropic effect. The entropy change represents about one-third of the total free energy barrier to the TSRC. Evidence from QM/MM calculations[51] and solvent KIEs (this work) suggest that the proton abstraction by Y149 proceeds ahead of the hydride transfer. Partial proton transfer is expected to cause release of electron density into the substrate C4-O bond and the consequent elongation of the C4-H bond will enable shortening of the DAD for efficient hydride transfer. Populating the TSRC from the GS thus involves conformational sampling and entropic penalty associated with it appears to correspond well with the idea of entropy funnel[41,46]. The S127A variant shows that precision of TSRC sampling can be perturbed massively without introducing energetic destabilization of the TSRC to an appreciable extent. The Y149F variant reflects large destabilization of the TSRC without introducing substantial change in GS preorganization compared to wild-type. The R88A variant introduces defect in enzyme preorganization, with a consequent destabilization of the TSRC that conformational sampling is unable to overcome. Overall, these results establish the requirement of rigid positioning at the TSRC for efficient C-H activation in an enzyme active site that requires conformational flexibility in fulfillment of its natural epimerase function.

## Methods

### Materials
Glucose, uridine 5′-triphosphate (UTP), α-D-glucose 1,6-bisphosphate and sodium pyruvate were from Sigma-Aldrich (Vienna, Austria). [4-²H]-glucose was from Omicron Biochemicals, Inc. (South Bend, USA). Deuterium oxide (99.96% ²H) was from Euriso-Top (Saint-Aubin Cedex, France). NAD⁺ was purchased from Roth (Karlsruhe, Germany). Hexokinase, α-phosphoglucomutase and D-lactate dehydrogenase were from Megazyme (Vienna, Austria). Other enzymes used for synthesis were prepared in-house. A GeneJET Plasmid Miniprep Kit (Thermo Scientific, Waltham, MA, USA) was used for plasmid DNA isolation. Q5® High-Fidelity DNA polymerase and DpnI were purchased from New England Biolabs (Frankfurt am Main, Germany). *E. coli* BL21(DE3) competent cells were prepared in-house. Oligonucleotide primers were sourced from Sigma-Aldrich. Chemicals and reagents were of highest available purity.

### Enzymatic synthesis of UDP-[4-²H]-GlcA
15 mM 4-[²H]-glucose (100 mg, 0.55 mmol), 50 mM uridine 5′-triphosphate (716.5 mg, 1.3 mmol), 0.13% (w/v) bovine serum albumin (48.1 mg), 5 mM MgCl₂ (17.6 mg, 0.18 mmol) and 10 μM α-D-glucose 1,6-bisphosphate were dissolved in 37 mL 50 mM Tris/HCl buffer (pH 7.5). The reaction was initiated by addition of hexokinase (13.8 U mL⁻¹, 510.6 U), α-phosphoglucomutase (251.6 U), inorganic pyrophosphatase from *E. coli* (2.8 mg, 0.077 mg mL⁻¹) and UDP-glucose pyrophosphorylase from *Bifidobacterium longum* (7.4 mg, 0.2 mg mL⁻¹). The mixture was incubated for 20 h at 30 °C without agitation. Oxidation to UDP-[4-²H]-GlcA by human UDP-glucose 6-dehydrogenase (1.23 mg mL⁻¹, 45.5 mg) was coupled to a co-enzyme regeneration system using 2 mM NAD⁺ (49 mg, 0.074 mmol), 40 mM sodium pyruvate (163 mg, 1.48 mmol) and D-lactate dehydrogenase (20 U mL⁻¹, 740 U). The reaction was incubated at 37 °C for 24 h. Enzymes were removed by ultrafiltration using Vivaspin centrifugal tubes (10 kDa cutoff, Sartorius) prior to product isolation. Synthesis of UDP-GlcA was conducted identically. Further description is provided elsewhere[48].

### Product isolation and characterization
UDP-GlcA and UDP-[4-²H]-GlcA were isolated in two consecutive steps. Product separation was accomplished using an ÄKTA FPLC system (GE Healthcare, Vienna, Austria) for anion exchange chromatography (125 mL Toyopearl SuperQ-650 M column; Tosoh Bioscience, Tokyo, Japan). An aqueous solution containing 20 mM sodium acetate (pH 4.3) was used as binding buffer. Compounds were eluted with a stepwise gradient by 1 M sodium acetate buffer (pH 4.3). Product containing fractions were detected by UV absorption ($\lambda$ = 254 nm), pooled and concentrated under reduced pressure (20 mbar, 40 °C) on a rotary evaporator (Laborota 4000, Heidolph, Schwabach, Germany). Sodium acetate was removed by gel filtration (Sephadex G-10 size exclusion column; GE Healthcare) with deionized water as eluent. The sugar nucleotide product was freeze dried on a Christ Alpha 1–4 lyophilizer (B. Braun Biotech International, Melsungen, Germany)[48]. UDP-GlcA and UDP-[4-²H]-GlcA were obtained as white powder with an isolated mass of ~130 mg (~40% yield). Each compound was ≥99.5% pure (HPLC; Supplementary Fig. 6) and the [²H] content at C4 was ≥99% (NMR; Supplementary Figs. 7 and 8).

### Enzyme variants
Wild-type UGAepi[48] and single-site variants thereof (R88A[50], S127A[51], Y149F[48]) were described elsewhere. Mutations to change P85 by glycine were made here (Supplementary Table 2). Purified enzymes were obtained by reported methods[48]. Briefly, expression was done in *E. coli* Lemo21 (DE3) cells, harboring pET17b vector with the relevant gene inserted. Proteins were carefully purified via Strep-tag II fused to their C-terminus. Purity was confirmed by SDS-PAGE (Supplementary Fig. 9 for P85G). The standard enzyme assay measured conversion of UDP-GlcA (4.0 mM) by HPLC[48]. Reactions were done in sodium phosphate buffer (50 mM, pH 7.6; 100 mM NaCl) at 25 °C without agitation.

### Determination of enzyme-bound NADH
NADH content in enzymes as isolated and during reaction at steady state was determined by reported methods[48]. Briefly, enzyme solution was incubated for denaturation with methanol (1:1, by volume) for 3 h at 25 °C without agitation (thermomixer comfort; Eppendorf AG). Mixture was centrifuged for 80 min at 16,100 × g (4 °C) to remove the enzyme precipitate, and the supernatant was isolated and analyzed on HPLC. Rapid-quench procedure[48] was used to measure enzyme-NADH during reaction (1 mL; 50 mM Na₂HPO₄ buffer, 100 mM NaCl, pH 7.6) with saturating concentration of UDP-GlcA (4.0 mM). Enzyme was used at 1.0–14 μM, depending on the activity. Reactions were conducted for 5 min at 25 °C (all variants) and 4 °C (S127A, P85G) followed by quenching with the same volume (1.0 mL) of ice-cold sodium phosphate buffer at pH 1.3. Mixtures were concentrated to 50 μL in VivaSpin tubes (30 kDa cutoff; 0 °C, 2880 × g), washed with 950 μL sodium phosphate buffer (pH 4.4), concentrated again to 50 μL, and then supplied with methanol (1:1, by volume) for enzyme denaturation. Sample work-up and analysis were done as described above. Values of mol% occupancy of enzyme with NADH are summarized in Supplementary Table 1.

### Enzyme stability
Melting temperature $T_m$. Wild-type UGAepi was buffered to 50 mM Na₂HPO₄, TRIS-HCl or MOPS (each pH 7.5) containing 50–250 mM NaCl. Enzyme solution (0.185 mg mL⁻¹; 5 μM; 45 μL) was mixed with 5 μL of 100-times diluted SYPRO ORANGE protein gel stain (Sigma-Aldrich). Samples were prepared in triplicates (50 μL total volume). A Bio-Rad CFX Connect™ Real-Time PCR Detection System was used. Changes in fluorescence intensity were detected using FRET mode (fast scan). The temperature ramp was set to 0.5 °C/min and melting curves recorded in a range of 20–99 °C (Supplementary Fig. 10, Supplementary Table 3). The lid temperature was 105 °C. The CFX Maestro software v1.0 (Bio-Rad) was used for data collection and analysis.

Enzyme inactivation. Sample (10 μL) taken at the end of incubation at 50 °C for initial rate analysis was diluted 10-fold into the standard activity assay and the residual enzyme activity was determined.

## Determination of $k_{cat}$ at variable temperature

Rate measurement. Initial reaction rates ($V$) were determined in sodium phosphate buffer (50 mM, pH 7.6; 100 mM NaCl) at variable temperature in the range 4–50 °C. The pH was verified at the temperature used and confirmed to be unchanged after the reaction. UDP-GlcA or UDP-[4-²H]-GlcA was used as substrate at 4.0 mM, equivalent to 11 times the $K_m$ of wild-type UGAepi at 25 °C. Enzyme solutions (25 μL) were equilibrated at each temperature in a non-agitated Eppendorf thermomixer comfort. Reactions were started with 5 μL substrate solution (24 mM). Samples (2.0 μL) were taken every 2.5 or 5.0 min, diluted 15-fold in $H_2O$ and quenched with methanol (30 μL). Mixtures were vortexed (10 s) and centrifuged (45 min, 21130 g) prior to HPLC analysis. The rate $V$ was determined from linear plot ($R^2 \geq 0.96$) of UDP-GalA product released with time (≤30 min). The $V$ values were typically recorded in the range 0.02–0.60 μM/sec. When working with the less active variants of UGAepi, the rate was adjusted by the enzyme concentration. Substrate conversion was below 10% in all experiments. Note: UDP-GlcA was cleanly converted to UDP-GalA in all reactions. The transient 4-keto-intermediate (Fig. 1a) or a decarboxylation product was not released in detectable amounts (≤0.01% of substrate converted). Measurements were performed in triplicates and typically agree within ±5.3%. Reactions at higher substrate concentrations (≥5.0 mM) showed $V$ unchanged within limits of error compared to reaction at 4.0 mM. The reported rates are therefore from conditions of enzyme saturation with substrate.

Determination of $k_{cat}$ and KIE on $k_{cat}$. The relationship $k_{cat} = V/[E]$ was used where [E] is the molar concentration of the enzyme subunit, determined with the molar extinction coefficient of UGAepi (36120 $M^{-1}$ $cm^{-1}$; 280 nm, 25 °C). The noncompetitive method was used to determine the KIE on $k_{cat}$. Individual reactions were performed to acquire initial rates with the ¹H and ²H isotope of UDP-GlcA. Superscript D ($^D k_{cat}$) indicates the KIE. The KIE was calculated with Eq. (3).

$$^D k_{cat} = k_{cat}(^1H)/k_{cat}(^2H) \qquad (3)$$

Determination of solvent kinetic isotope effects (SKIE). Enzyme (wild-type, Y149F) was buffered to sodium phosphate-²H₂O (50 mM, p²H 8.0, 100 mM NaCl) by repeated centrifugation (2100 × g, 4 °C) in 0.5 mL VivaSpin tubes (10 kDa cut-off) until a dilution factor of 400 was reached. The enzyme was incubated at 4 °C for 0.5 h to ensure that the hydron exchange was sufficient. No loss of enzyme activity (measured in ¹H₂O-buffer) was detected in the process. UDP-GlcA stock solutions were prepared in sodium phosphate-²H₂O (50 mM). Enzymes stored in sodium phosphate-¹H₂O buffer (50 mM Na₂HPO₄, 100 mM NaCl) were used for reactions in ¹H₂O. SKIE measurements were performed at 25 °C and pL 8.0 (L = ¹H or ²H). The p²H was obtained as pH meter reading +0.4. Note that the $k_{cat}$ of UGAepi is insensitive to pH change in the range 6.5–9.0[48]. An observable SKIE could therefore not arise from a different ionization of the enzyme-substrate complex in ²H₂O compared to ¹H₂O. Viscosity controls[73,74] were performed in 9% v/v glycerol prepared in sodium phosphate-¹H₂O (50 mM, pH 8.0) and 100 mM NaCl. The $k_{cat}$ of both enzymes (wild-type, Y149F) was unaffected by the glycerol microviscogen (±4%). Reaction setup and sampling for determination of $V$ were identical to the procedure described above. SKIEs were calculated as the ratio of the $k_{cat}$ in ¹H₂O and ²H₂O (Eq. (4)).

$$^{D2O} k_{cat} = k_{cat,L}(^1H_2O)/k_{cat,L}(^2H_2O) \qquad (4)$$

Subscript L indicates the ¹H or ²H isotope of substrate.

Temperature profiles of $k_{cat}$. Curved temperature dependence of ln $k_{cat}$ vs. $T$ was fitted with Eq. (2). Nonlinear fitting was done with SigmaPlot 15.0 (Systat Software, Inc.).

## HPLC

A Shimadzu Prominence HPLC-UV system equipped with a Kinetex® C18 analytical HPLC column (50 × 4.6 mm, 5 μm 100 Å; Phenomenex, Aschaffenburg, Germany) was used. UV-detection was set to 262 nm and injection volumes of 5.0–30 μL were used. UDP-sugars, NAD⁺ and NADH were separated applying an isocratic flow (2 mL/min) at 40 °C with a mobile phase composed of 20 mM potassium phosphate buffer (pH 5.9) containing 40 mM tetrabutylammonium bromide (95%; solvent A) and acetonitrile (5%; solvent B).

## ¹H NMR

Data acquisition was carried out on a Varian INOVA 500-MHz NMR spectrometer (Agilent Technologies, Santa Clara, California, USA). The VNMRJ 2.2D software was used for measurements. ¹H NMR spectra (499.98 MHz) were recorded in D₂O on a 5 mm indirect detection PFG-probe with pre-saturation of the water signal by a shaped pulse. Spectra were analyzed using MestReNova 16.0 (Mestrelab Research, S.L., Santiago de Compostela, Spain).

## Protein structures

PyMOL Molecular Graphics System 2.5.4 (Open-Source, Schrödinger, LLC) was used for structural depictions.

## Reporting summary

Further information on research design is available in the Nature Portfolio Reporting Summary linked to this article.

## Data availability

All data are available from the corresponding author (B.N.) upon request. The authors declare that all data supporting the findings of this study are available within the article and its Supplementary Information. Additionally, data for Figs. 2, 3 and Supplementary Fig. 9 are provided in the Source Data file. All crystal structures used in this study are available in the RCSB Protein Data Bank under following PDB identifiers: 6ZLL, 6ZLD, 6ZLJ. Source data are provided with this paper.

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

## Acknowledgements

Financial support from the Austrian Science Fund (FWF; projects n° I 3247 EpiSwitch and n° I 4516 DeoxyBioCat, B.N.) is gratefully acknowledged.

## Author contributions

C.R., design of study, experiments and data analysis, writing of paper; A.B., experiments and data analysis; B.N., design of study, funding acquisition, discussion, writing of paper.

## Competing interests

The authors declare no competing interests.
