## [Peer Review File · Nature Communications]

Interplay of structural preorganization and conformational sampling in UDP-glucuronic acid 4-epimerase catalysisEditorial Note: This manuscript has been previously reviewed at another journal that is not operating a transparent peer review scheme. This document only contains reviewer comments and rebuttal letters for versions considered at *Nature Communications*.

REVIEWER COMMENTS

Reviewer #3 (Remarks to the Author):

My main concern were taken into account and the paper can be published

Reviewer #4 (Remarks to the Author):

This manuscript presents some very interesting results of wide importance and interest. The referees have worked hard to revise the paper in line with the referees' extensive comments and I appreciate in particular their balanced and considered description of the proposals and results throughout. However, as noted by the referees, there are some issues and problems around the analysis of the results, and their presentation. Overall, the modifications of the text and the presentation only partially address the questions and points raised by the referees (see below). In my view the importance of the results for a topical area of wide debate means that they deserve publication in a high impact journal such as *Nature Communications*. The manuscript itself requires revision as noted by all referees, some of whose comments apply to the revised version.

The results are an important contribution in the area of the temperature dependence of enzyme catalysed reactions, but the discussion is flawed. The authors' experimental results show interesting temperature dependence of the chemical reaction rate in an enzyme. The questions raised by the referees include the potentially problematic use of a phenomenological model to analyse the results in terms of donor acceptor distance and distinct populations of the enzyme reactant complex assumed to react either classically or by quantum tunnelling, respectively. As the kinetic isotope effects show, the contribution to reaction from tunnelling here is small, as expected for this type of enzyme. Therefore, a model focused on quantum tunnelling is not really appropriate for analysis. Similarly, the

use of the term "tunnelling ready state" is not appropriate; instead, a more general term such as "transition state like conformation" should be used. The model used originally developed by Kohen et al. is potentially useful in some limited cases but does not provide a direct physical interpretation. Rather, it is a two-state model (of which there are many in the literature) of which the interpretation of the parameters is open to debate. For direct interpretation, molecular simulations are required as the authors note. They also note that the model they present does not provide a precise fit, with some parameters (such as the distance R_0) clearly uncertain, an indication that it is not adequate here (see also comments of Reviewer 2). If the model fitting is to be retained (and the case for doing so is not strong), it should be moved to supporting information and appropriate caveats should be added to the discussion. The authors point out some relevant molecular simulation (eg QM/MM) work on other enzymes, which can provide molecular, atomic-level insight. So, I do not find the title of the paper justified.

I also note that to correctly capture tunnelling and other contributions in a fitting model, the temperature dependence of tunnelling must also be taken into account see for example, Glowacki et al. *Nature Chemistry* 4, 169-176 2012; Truhlar J. *Phys. Org. Chem* 23, 660-676 2010.

However to be clear, the experimental results in the manuscript are interesting and important and worthy of publication in high impact journal. The experiments have been performed carefully and tested appropriately. The central important findings are well defined temperature dependence of kinetic isotope effects and, particularly, the finding of significant activation heat capacities. The authors show nicely that different activation heat capacities found for different mutants, leading to significantly different temperature independent behaviour. This is important: activation capacity effects are the focus of much debate in the current literature and these findings are an important contribution. The authors describe much of this relevant background. They show that the results are fit well by equation 3, which they describe as the heat capacity model. This is Macromolecular Rate Theory (MMRT), proposed and detailed by Arcus et al. *Biochemistry* 55, 1681-1688 (2016), and should be described as such. The authors do cite other relevant references by Arcus et al. The differences in activation heat capacities between the mutants are intriguing and in

particular how they relate to: firstly, differences in catalytic activity; and secondly, differences in temperature dependent behaviour and optimum temperatures.

This has crucial implications also for the interpretation of the results. The authors focus on analysing activation enthalpies and entropies at a particular temperature. However, as the authors themselves note in the discussion< the existence of an activation heat capacity means that both the activation entropy on the activation enthalpy are temperature dependent. Both flow from the activation heat capacity, as does the activation free energy. The fundamentally important quantity, as indeed in studies of protein folding, is the heat capacity. Explanations of activity based on entropy and enthalpy are therefore only valid at one specific temperature and not satisfactory as a general explanation. Entropy is overemphasised in the manuscript. If the authors wish to focus on entropy and entropy, they must make clear that their arguments apply only to one specific temperature. They do include this in part in their discussion, but this point is lost in the focus on entropic effects. This relates to questions raised by the referees and requires revision throughout the text. The arguments based on entropy (and entropy "funnels") are not general, as the data show

Fits to the MMRT model are shown in Figure S2, for the strongly and weakly temperature-dependent mutants. These fits should be shown in the main text in addition to the raw data shown in figure 3. The results show an interesting breakpoint behaviour. As the authors note, the data show two different regimes. This can be accounted for by the recent developments of Arcus et al. (Walker et al. bioRxiv 2023.07.06.548038; doi: <https://doi.org/10.1101/2023.07.06.548038>): they show that such behaviour is accounted for by the different heat capacities of different forms of the enzyme substrate complex. Specifically, the conformation that dominates reaction, the transition state like conformation, exists in equilibrium with other forms all the enzyme substrate ensemble and is expected to have a lower heat capacity. The shifting population of the TLC accounts for the behaviour observed: this is two-state MMRT. I would ask the authors to test the fitting of their data by this model, and predict it will explain their observations. This also relates to questions about alcohol dehydrogenase asked by referees: two state MMRT also accounts for the temperature dependence of ADH activity, as discussed by Arcus et al.

More minor points

On Page 3 the authors stated that Klinmann and coworkers introduced the idea of an entropy penalty and its relation to catalysis. Such proposals predate that work, for example the so-called Circe hypothesis of Jencks (*Adv. Enzymol. Relat. Areas Biochem.* 43: 219–410 (1975), and his *Catalysis in Chemistry and Enzymology* textbook, 1987). Entropy effects in enzyme reactions have been analysed by Aqvist et al. *Acc. Chem. Res.* 2017, 50, 2, 199–207) and Warshel et al. *Proc Natl Acad Sci USA.* 2000 97:11899-904. As a general point, entropy effects can be important in determining temperature dependence (when activation heat capacities are small, see above), but are not very important in catalysis, i.e. they are not important in rate acceleration relative to an equivalent uncatalyzed reaction, shown by Aqvist and Warshel. (Also, tunnelling can contribute significantly to determining the reaction rate in some enzymes (though not in this case), but still not contribute significantly to catalysis (e.g. Warshel 2009 <https://doi.org/10.1039/9781847559975>; *J. Phys. Org. Chem* 23, 677-684, 2010)). This leads to the relatively minor but important point is that I would ask the authors to be careful about the use of the term catalysis: strictly, of course catalysis means acceleration of a reaction, and its casual use to mean 'reaction' or 'activity' should be avoided to avoid confusion. Hence on page 5, please change "measurements to probe the energetics and conformational characteristics of UGAepi catalysis" to "measurements to probe the energetics and conformational characteristics of UGAepi activity", and similarly elsewhere, including in subsection headings.

Page 9: "computational studies" is imprecise: molecular simulations should be distinguished from other computational approaches.

Page 13: Conformational effects can be affected by deuteration (e.g. through changes in hydrogen bond strength).

Also relevant in the discussion of the effects of the iteration is work on comparing heavy (isotopically substituted) and light DHFR (Luk et al. *PNAS* 110, 16344-16349 (2013), which showed that 100% substitution of the whole enzyme leads to a small but measurable dynamical effect on reaction. This is not due to a change in quantum tunnelling (which is in any case not a large contributor to the reaction, similar to the present case) but rather reflects effective changes in viscosity and dynamical recrossing in the heavy enzyme.

Another minor point is that I do not understand the use of the three letter acronym GST for the ground state: simply GS seems simpler and less ambiguous to me.

In reporting the changes in activation heat capacity, is it justified to report these to two decimal places? (e.g. $\Delta\Delta C_p^\ddagger = 3.42 \text{ kJ/mol.K}$, pg. 19).

Responses to referees.

Reviewer one raises some eccentric objections. I do not agree this work is old fashioned and there is a need for detailed kinetic studies to resolve questions of temperature dependence of enzyme catalysed reactions including the study of kinetic isotope effects. The authors rebut this effectively. The referee also questions the application of thermodynamics in understanding catalysis regarding it as simplistic. Again the authors respond well to this. I note that investigations of nonbiological catalysts and of other biological processes such as protein folding also successfully apply thermodynamic arguments and principles. The authors provide to me convincing evidence that the mechanism does not change over the temperature range studied and that they are indeed investigating the crucial chemical step, which is why the results are convincing and important. However, there is one valid criticism made by this reviewer that is not adequately addressed in the revised manuscript: that is the limitations of the simple donor-acceptor model referred to above. I agree with the referee that such fitting is fraught with potential misinterpretation and if this simple model is to be included in the manuscript full discussion of its limitations and potential alternative interpretations should be given.

Reviewer 2 raises the interesting case of thermophilic ADH to which I refer above and is also relevant to the comments of reviewer 3. The recent work of Arcus et al. on 2-state MMRT is relevant here as noted above in their response to the reviewer, the authors focus on entropy contributions but as discussed above, and shown by their data, these arguments are not general and only apply at one specific temperature. The finding of an activation heat capacity shows that different activation entropies will be found at different temperatures. The discussion needs to be refocused throughout (abstract, discussion and conclusions) to correctly reflect and describe the experimental observations reported here. This reviewer also notes limitations in the simple Marcus type model used, and the fact that it does not

fully describe the observed behaviour for all mutants. This relates to points discussed above and the discussion in the money script should include limitations and potential misinterpretations of the model (if it is to be retained in the manuscript) as discussed above. The discussion of interatomic distances based on this model is not justified as the data and the model do not permit these distances to be identified with the level of accuracy and precision stated. Regarding the referee's point about Aqvist's proposals on changes in rate-limiting step and inactive forms of the enzyme, the authors do a good job in rebutting those possibilities undemonstrated that their data do indeed refer to the chemical step and are not complicated by conformational complexity. The question about water molecules is an interesting one and the authors deal well with that in their discussion. It is interesting to discuss what stabilises the tyrosinate anion.

For reviewer 3: the authors answer most points well and have made useful and effective revisions. As noted above the turn of the discussion in the manuscript is sensible and the language is clear. It is well written. On the question of entropy and tunnelling raised by this reviewer 3, the revisions do not fully satisfy the questions raised in my view. Entropy effects should be clarified as temperature dependent, and the effect on catalysis of tunnelling and entropy should be made clear. The response repeats the confusion between catalysis and reaction: k_{cat} measures the rate of reaction, it does not measure catalysis, which is the acceleration of the reaction rate. The need for clear distinction between reactivity and catalysis is discussed above and has been discussed by Warshel et al. The manuscript should be modified to avoid such confusion

REVIEWER COMMENTS

Reviewer #3 (Remarks to the Author):

My main concern were taken into account and the paper can be published

Response: No more changes were necessary in response to this Reviewer.

Reviewer #4 (Remarks to the Author):

This manuscript presents some very interesting results of wide importance and interest. The referees have worked hard to revise the paper in line with the referees' extensive comments and I appreciate in particular their balanced and considered description of the proposals and results throughout. However, as noted by the referees, there are some issues and problems around the analysis of the results, and their presentation. Overall, the modifications of the text and the presentation only partially address the questions and points raised by the referees (see below). In my view the importance of the results for a topical area of wide debate means that they deserve publication in a high impact journal such as Nature Communications. The manuscript itself requires revision as noted by all referees, some of whose comments apply to the revised version.

Response: We thank the Reviewer for the generally positive evaluation of our manuscript. Points raised by Reviewer 4 were addressed by revision as described below in full detail. The comments of Reviewer 4 were not organized in an itemized list. To avoid that we lose important points raised, I highlight statements that were specifically addressed in our response.

The results are an important contribution in the area of the temperature dependence of enzyme catalysed reactions, **but the discussion is flawed**. The authors' experimental results show interesting temperature dependence of the chemical reaction rate in an enzyme. The questions raised by the referees include the **potentially problematic use of a phenomenological model** to analyse the results in terms of donor acceptor distance and distinct populations of the enzyme reactant complex assumed to react either classically or by quantum tunnelling, respectively. As the kinetic isotope effects show, the contribution to reaction **from tunnelling here is small**, as expected for this type of enzyme. Therefore, a model focused on quantum tunnelling is not really appropriate for analysis. Similarly, the use of the term **"tunnelling ready state" is not appropriate; instead, a more general term such as "transition state like conformation" should be used**. The model used originally developed by Kohen et al. is potentially useful in some limited cases **but does not provide a direct physical interpretation**. Rather, it is a two-state model (of which there are many in the literature) of which the interpretation of the parameters is open to debate. For direct interpretation, molecular simulations are required as the authors note. They also note that the model they present does not provide a precise fit, with some parameters (such as the distance R_0) clearly uncertain, an indication that it is not adequate here (see also comments of Reviewer 2). If the model fitting is to be retained **(and the case for doing so is not strong)**, it should be moved to supporting information and appropriate caveats should be added to the discussion. The authors point out some relevant molecular simulation (eg QM/MM) work on other enzymes, which can provide molecular, atomic-level insight. So, **I do not find the title of the paper justified**.

Response/changes: We agree with the Reviewer in general. We do not have support for the phenomenological model of Kohen et al except that it was used previously with relevantly similar nicotinamide coenzyme-dependent enzymes (e.g., alcohol dehydrogenase, dihydrofolate reductase). The model is not clear in the parameters involved, particularly the energy difference between enzyme populations is debatable in its physical meaning. As solution, we propose to omit everything related to this model. The manuscript has been revised accordingly. The fitting results were removed (see Table 1). The immediate text parts were also removed, of course. The overall manuscript text was changed to avoid focus on donor-acceptor distances that, we agree, was too strong given the evidences provided. More specifically regarding detailed points raised in the paragraph of Reviewer comments above (highlighted in yellow for facile retrieval), we avoid use of the term “tunneling ready state (TRS)” and replace it by the term “transition state-like reactive conformation (TSRC)”. We considered adopting the nomenclature of “transition state-like conformation (TLC)” of Walker et al from the reference mentioned by the Reviewer below. However, for experimentalists, the abbreviation of TLC is reserved to mean thin-layer chromatography. Additionally, the concept of Walker et al seems to imply a transition state-like conformation that still requires a substantial amount of energy to proceed to the actual transition state (see the Graphical Abstract of the paper now appeared in ACS Catalysis). This is *not* what we have in mind, hence the use of the word “reactive” in TSRC. We agree with the Reviewer on the criticism regarding the contribution from quantum-mechanical tunneling. Besides removing the model to fit the temperature profiles of KIEs, we have been careful in writing in general. Point is that we simply don’t know the contribution from tunneling in this reaction. It may be small as the Reviewer suggests. The relatively small KIEs would certainly support this view. In an earlier study involving some of the authors, QM/MM calculations in molecular simulations on the epimerase reaction were done classically (excluding tunneling effects). The title of the paper was changed to remove the focus on donor-acceptor distances. A point raised later is that on the use of the word “catalysis”. We agree with Reviewer 4 in general and have replaced terminology at various where, indeed, the use of catalysis was not precise. In the title, however, we find the use of catalysis to have been appropriate. After all, the catalysis of an enzyme often involves elements of conformational change, especially stochastic sampling of the reactive conformation. Although conformational sampling does not immediately contribute to catalysis in the sense of rate acceleration, as the Reviewer points out later and we agree, it is part of the catalytic process in the enzyme.

I also note that to correctly capture tunnelling and other contributions in a fitting model, the temperature dependence of tunnelling must also be taken into account see for example, Glowacki et al. Nature Chemistry 4, 169-176 2012; Truhlar J. Phys. Org. Chem 23, 660-676 2010.

Response/changes: We agree with the Reviewer. Fit of the data with the phenomenological model for tunneling raises more questions/concerns that it helps in interpreting the experimental results. As mentioned above, everything related to the model fit was removed.

However to be clear, the experimental results in the manuscript are interesting and important and worthy of publication in high impact journal. The experiments have been performed carefully and tested appropriately. The central important findings are well defined temperature dependence of kinetic isotope effects and, particularly, the finding of significant activation heat capacities. The authors show nicely that different activation heat

capacities found for different mutants, leading to significantly different temperature independent behaviour. This is important: activation capacity effects are the focus of much debate in the current literature and these findings are an important contribution. The authors describe much of this relevant background. They show that the results are fit well by equation 3, which they describe as the heat capacity model. This is **Macromolecular Rate Theory (MMRT)**, proposed and detailed by Arcus et al. *Biochemistry* 55, 1681-1688 (2016), and should be described as such. The authors do cite other relevant references by Arcus et al. The differences in activation heat capacities between the mutants are intriguing and in particular how they relate to: firstly, differences in catalytic activity; and secondly, differences in temperature dependent behaviour and optimum temperatures.

Response: We thank the Reviewer for the positive assessment of the experimental part of our study. The term Macromolecular Rate Theory (MMRT) is mentioned and the additional literature cited. As the Reviewer points out, we did mention and cite the development of MMRT, including a major review by the people mainly responsible for the development.

This has crucial implications also for the interpretation of the results. The authors focus on analysing activation enthalpies and entropies at a particular temperature. However, as the authors themselves note in the discussion< the existence of an activation heat capacity means that both the activation entropy on the activation enthalpy are temperature dependent. Both flow from the activation heat capacity, as does the activation free energy. The fundamentally important quantity, as indeed in studies of protein folding, is the heat capacity. Explanations of activity based on entropy and enthalpy are therefore only valid at one specific temperature and not satisfactory as a general explanation. **Entropy is overemphasised in the manuscript. If the authors wish to focus on entropy and entropy, they must make clear that their arguments apply only to one specific temperature.** They do include this in part in their discussion, but this point is lost in the focus on entropic effects. This relates to questions raised by the referees and requires revision throughout the text.

The arguments based on entropy (and entropy "funnels") are not general, as the data show

Response/changes: We agree with the Reviewer, of course, that enthalpies and entropies are temperature dependent for the epimerase reaction. It is what the evidence of our study implies. However, we do not agree that this fact invalidates discussion of the thermodynamic parameters which must be done, of course, at one specified temperature. We realize that we have been not clear enough on the point of fixed temperature used. We address the point of the Reviewer in two ways. First, we include a new figure (Figure S2) in the Supplementary Information that shows the contribution from enthalpy and entropy to the overall Gibbs free energy of activation in a relevant* temperature range (293 – 303 K). [*Relevant as used in the study. We could extend relevance to physiological temperature of organism etc]. It becomes clear from this figure that in the reaction of the wild-type enzyme, the change in contribution from enthalpy and entropy is quite small. The contribution from entropy is substantial. In the reaction of the S127A variant, the change in relative contribution from enthalpy and entropy is larger, as expected from differences in the activation heat capacity for the reaction of this variant and the reaction of the wild-type. However, in both enzymatic reactions entropy is a significant factor of the overall ΔG and so it is, in our opinion, fully justified for discussion. While we make clear that our arguments are for the temperature specified (298 K), it is also implicit from Figure S2 that the general argument holds in principle for a broader temperature range. The detailed argument must be made for a defined temperature, as the Reviewer mentions. We agree.

Given the significant contribution from entropy to the overall Gibbs free energy, the authors reserve the right to ask in their paper where this entropic effect originates from. We suggest that the requirement for conformational sampling, to populate the TSRC from the GS, is a plausible interpretation. It seems reasonable to assume/consider that the conformational sampling will also include in some degree the search (“sampling”) of suitable DAD for reaction. As mentioned above, we have tempered our discussion on DAD and have eliminated all results from model fit. Concern of the Reviewer about whether this interpretation (“the argument”) on entropic effects is general seems unjustified in our opinion. It is also not completely clear to us what the term “general” refers to here. Does it mean that the argument does not hold over the whole temperature range examined or over a certain range of temperatures? Or is it “general” in a broader sense, like other enzymes? We show (see the new Figure S2) that temperature change (293 – 303 K) does not change the essence of the argument on entropy, which is discussed for a fixed temperature of 298 K. Data in Table 1 (k_{cat} , KIE, thermodynamic parameters) are shown for that temperature. We haven’t made any further generalization. Our position is that the role of entropy in the reaction of the epimerase, at the main temperature of measurement (298 K) or within the specifically analyzed temperature range (new Figure S2), deserves consideration and interpretation. This has also been a response to Reviewer 3 of the original manuscript and it will be noted that Reviewer 3 agreed with our response.

Fits to the MMRT model are shown in Figure S2, for the strongly and weakly temperature-dependent mutants. **These fits should be shown in the main text in addition to the raw data shown in figure 3.**

Response/changes: Done as requested, Figure 3 of main text shows the fits.

The results show an interesting breakpoint behaviour. As the authors note, the data show two different regimes. **This can be accounted for by the recent developments of Arcus et al. (Walker et al. bioRxiv 2023.07.06.548038; doi: <https://doi.org/10.1101/2023.07.06.548038>): they show that such behaviour is accounted for by the different heat capacities of different forms of the enzyme substrate complex.** Specifically, the conformation that dominates reaction, the transition state like conformation, exists in equilibrium with other forms all the enzyme substrate ensemble and is expected to have a lower heat capacity. The shifting population of the TLC accounts for the behaviour observed: this is two-state MMRT. I would ask the authors to test the fitting of their data by this model, **and predict it will explain their observations.** This also relates to questions about alcohol dehydrogenase asked by referees: two state MMRT also accounts for the temperature dependence of ADH activity, as discussed by Arcus et al.

Response/changes: We discuss on page 12 of our revised manuscript the new evidence from the Walker et al paper just published in ACS Catalysis, in the extent we find it relevant for our current study. We also show in Figure S3 (Supplementary Information) the results of our attempts to use the new MMRT model for the fitting of temperature-rate profiles (i.e., the dependence of the activation free energy ΔG on temperature). It is perhaps as expected: our data sets are too small for the model or the model is overparameterized for the data. However, the fits could be made to converge to a solution and the parameter values appear to be physically meaningful. However, the estimated value of the heat capacity change (ΔC_p) appears to be unrealistically high and it is positive (which is problematic, of course). We show a table (panel d of Figure S3) with the results obtained for wild-type, S127A and P85G. A short discussion is also provided in the Supplementary Information. In the main text, we

also comment on the interpretation of the “transition state-like conformation” (terminology of Walker et al.) as a conformational state populated on the way to the actual transition state.

Overall, we conclude therefore that the “standard” MMRT model of Arcus et al is fully sufficient to describe the continuous temperature dependencies of k_{cat} (or: the dependence of ΔG on temperature) for the different epimerases. While quite interesting as such, the expanded MMRT model (Walker et al.) has no immediate use for us at this moment, in the sense that fitting this model would provide deepened insight into the epimerase reaction. The Reviewer mentions that the expanded MMRT model might “explain our observations”. We assume that the Reviewer refers to Figure 3 (panel d) for the reaction of the P85G variant. This enzyme involves an unusual breakpoint in the Arrhenius plot of the k_{cat} . Fits of the data with the expanded (simplified) MMRT model are shown in Figure S3. The model can fit temperature profile of ΔG on temperature. The model involves an empirical parameter m that approximates the temperature dependence of the heat capacity term. However, there are various models that can describe the change at the level of the k_{cat} . We refer the Reviewer to the statement (used already in the original manuscript), preceding the discussion on the expanded MMRT model on page 12. The statement reads (in black font):

The molecular origin of discontinuity in the k_{cat} temperature profile of P85G was not pursued. It is not due to change of rate-limiting step, evidenced by absence of enzyme-NADH at steady state. Studies of other enzymes have explained similar breakpoints in temperature-rate profiles to arise from a temperature-dependent change in the equilibrium population of protein conformational sub-states (effectively f_{R} , see Equation 1).^{29,56,68,69}

In our opinion, this statement has it all and there is nothing of important substance that additional fits with the expanded MMRT model are able to add here. Multiple models fit the data and generally do so with similar quality (or difficulty). Models cannot be discriminated reliably on an objective basis. Note our cautious efforts to exclude, based on independent evidence, the Aquist model to explain the development of curvature in Arrhenius plots. Fits of more complex models won't be the solution. The study of Walker et al is strong in that molecular simulations provide independent support to the use of certain mathematical models of temperature dependence. Here, we cannot rely on molecular simulation. We therefore use the simplest model consistent with the data. In previous work (reference 56), we explored with more data than we have here for P85G, the breakpoint in the Arrhenius plot of k_{cat} for a thermophilic CDP-glucose 2-epimerase. At this stage, we can say that the MMRT model fits the data for P85G but so would other models. We refer to the sentence marked in yellow above.

The Reviewer also mentions alcohol dehydrogenase. It is important to note that the temperature dependence of the thermophilic ADH is different from the temperature dependence of UGAepi (or variants thereof). In terms of thermodynamic activation parameters, difference is in the contribution of entropy to the overall Gibbs free energy of activation. The simple fact that the expanded MMRT model is able to fit the data for the alcohol dehydrogenase does not mean that the model can be generalized. As mentioned above, there are other mathematical/kinetic models to fit the same temperature dependencies.

More minor points

On Page 3 the authors stated that Klinmann and coworkers introduced the idea of an entropy penalty and its relation to catalysis. Such proposals predate that work, for example the so-called Circe hypothesis of Jencks (Adv. Enzymol. Relat. Areas Biochem. 43: 219–410

(1975), and his Catalysis in Chemistry and Enzymology textbook, 1987). Entropy effects in enzyme reactions have been analysed by Aqvist et al. *Acc. Chem. Res.* 2017, 50, 2, 199–207) and Warshel et al. *Proc Natl Acad Sci USA.* 2000 97:11899-904.

Response/changes: We agree with the Reviewer and have included the point in the Introduction, citing the relevant papers, especially Jencks and Aqvist. The position of Warshel and co-workers is discussed at different places in the manuscript.

As a general point, entropy effects can be important in determining temperature dependence (when activation heat capacities are small, see above), but are not very important in catalysis, i.e. they are not important in rate acceleration relative to an equivalent uncatalyzed reaction, shown by Aqvist and Warshel. (Also, tunnelling can contribute significantly to determining the reaction rate in some enzymes (though not in this case), but still not contribute significantly to catalysis (e.g. Warshel 2009 <https://doi.org/10.1039/9781847559975>; *J. Phys. Org. Chem* 23, 677-684, 2010)). This leads to the relatively minor but important point is that I would ask the authors to be careful about the use of the term catalysis: strictly, of course catalysis means acceleration of a reaction, and its casual use to mean 'reaction' or 'activity' should be avoided to avoid confusion. Hence on page 5, please change "measurements to probe the energetics and conformational characteristics of UGAepi catalysis" to "measurements to probe the energetics and conformational characteristics of UGAepi activity", and similarly elsewhere, including in subsection headings.

Response/changes: We agree with the Reviewer. Changes ("catalysis" substituted by the appropriate term, e.g., "activity", "reaction") were made in various places of the manuscript, including the titles of the sub-sections mentioned. In the manuscript title, however, we suggest that "catalysis" is the correct term.

Page 9: "computational studies" is imprecise: molecular simulations should be distinguished from other computational approaches.

Response/changes: Terminology suggested by the Reviewer was adopted throughout.

Page 13: Conformational effects can be affected by deuteration (e.g. through changes in hydrogen bond strength).

Response: This is certainly true, but we are not aware of changes in protein conformation caused by the isotopic substitution of the hydrogen in substrate, in particular a hydrogen transferred as hydride in an oxidoreductase reaction. The reactive hydrogen does not engage in a typical/classical hydrogen bond. Our argument that substrate deuteration is exceedingly unlikely to affect temperature dependence via (global) effect on protein conformation remains.

Also relevant in the discussion of the effects of the iteration is work on comparing heavy (isotopically substituted) and light DHFR (Luk et al. *PNAS* 110, 16344-16349 (2013)), which showed that 100% substitution of the whole enzyme leads to a small but measurable dynamical effect on reaction. This is not due to a change in quantum tunnelling (which is in any case not a large contributor to the reaction, similar to the present case) but rather reflects effective changes in viscosity and dynamical recrossing in the heavy enzyme.

Response: We agree with the Reviewer on the effect of isotopic substitution in the enzyme. However, (global) labeling of protein with isotopes differs in many relevant respects from the substitution of a single hydrogen site by deuterium in the substrate. We are unable to

see the immediate connection between the two types of isotopic labeling, despite the fact that both have been used in the study of protein dynamics related to enzyme catalysis. An expanded discussion to include effects of isotope labeling of the enzyme seems to be unnecessary, potentially misleading even. In our opinion, these are two completely different things that only have in common that natural isotopes are substituted. Only as a reminder, the argument is that deuteration of substrate is unlikely to cause change in the temperature dependence of k_{cat} due to global effect on protein conformation (e.g., the substrate-bound enzyme is less stable when 4'-deuterated UDP-GlcA is used instead of normal UDP-GlcA).

Another minor point is that I do not understand the use of the three letter acronym GST for the ground state: simply GS seems simpler and less ambiguous to me.

Response/changes: We agree, the abbreviation was changed.

In reporting the changes in activation heat capacity, is it justified to report these to two decimal places? (e.g. $\Delta\Delta C_p^\ddagger = 3.42 \text{ kJ/mol.K}$, pg. 19).

Response/changes: We agree, the decimal places are reduced to only one. In Table 1, the precision (two decimal places) is supported by the standard deviation from the non-linear fit of the temperature profiles.

Responses to referees.

Reviewer one raises some eccentric objections. I do not agree this work is old fashioned and there is a need for detailed kinetic studies to resolve questions of temperature dependence of enzyme catalysed reactions including the study of kinetic isotope effects. The authors rebut this effectively. The referee also questions the application of thermodynamics in understanding catalysis regarding it as simplistic. Again the authors respond well to this. I note that investigations of nonbiological catalysts and of other biological processes such as protein folding also successfully apply thermodynamic arguments and principles. The authors provide to me convincing evidence that the mechanism does not change over the temperature range studied and that they are indeed investigating the crucial chemical step, which is why the results are convincing and important. However, there is one valid criticism made by this reviewer that is not adequately addressed in the revised manuscript: that is the limitations of the simple donor-acceptor model referred to above. I agree with the referee that such fitting is fraught with potential misinterpretation and if this simple model is to be included in the manuscript full discussion of its limitations and potential alternative interpretations should be given.

Response/changes: As described above in response to the comments of Reviewer 4, the DAD model was removed.

Reviewer 2 raises the interesting case of thermophilic ADH to which I refer above and is also relevant to the comments of reviewer 3. The recent work of Arcus et al. on 2-state MMRT is relevant here as noted above in their response to the reviewer, the authors focus on entropy contributions but as discussed above, and shown by their data, these arguments are not general and only apply at one specific temperature. The finding of an activation heat capacity shows that different activation entropies will be found at different temperatures. The discussion needs to be refocused throughout (abstract, discussion and conclusions) to correctly reflect and describe the experimental observations reported here. This reviewer also notes limitations in the simple Marcus type model used, and the fact that it does not fully describe the observed behaviour for all mutants. This relates to points discussed above

and the discussion in the money script should include limitations and potential misinterpretations of the model (if it is to be retained in the manuscript) as discussed above. The discussion of interatomic distances based on this model is not justified as the data and the model do not permit these distances to be identified with the level of accuracy and precision stated. Regarding the referee's point about Aqvist's proposals on changes in rate-limiting step and inactive forms of the enzyme, the authors do a good job in rebutting those possibilities undemonstrated that their data do indeed refer to the chemical step and are not complicated by conformational complexity. The question about water molecules is an interesting one and the authors deal well with that in their discussion. **It is interesting to discuss what stabilises the tyrosinate anion.**

Response/changes: To begin with, the tunneling model was removed. Position of the authors with respect to ADH and expanded MMRT model has been described in the responses to Reviewer 4. Briefly, a series of papers describe the breakpoint in the temperature dependence of k_{cat} of ADH. This is however not our work. In respect to the P85G variant, we mention, with citations of the relevant papers on ADH, that the breakpoint phenomenon has been observed with other enzymes but that we have not pursued it in the current study. We have responded to the point about temperature dependence of enthalpy and entropy. We think it is fine to not repeat this here again. However, entropy plays a significant role in the epimerase reaction, as shown with evidence that was not under debate in this whole review process. In is our opinion, that the point about entropy needs to be emphasized properly. It is part of the identity of this study and an interpretation of the entropic effects must be given. We have in the Abstract revised the statement "Collectively, our study captures **thermodynamic** (previously: entropic) effects associated with TSRC sampling ..." to make clear that it isn't just about entropy.

Lastly, the stabilization of a tyrosinate anion is quite well established in this class of enzymes (short-chain dehydrogenases/reductases). The point is discussed in more detail in earlier references on this enzyme, especially reference #47. Discussion of the ionization behavior of the tyrosine in the current manuscript would open up a completely new direction/line of thought for the discussion that is however not necessary. We feel this would be rather be a distraction from the main lines of discussion, potentially leading to confusion of the reader.

For reviewer 3: the authors answer most points well and have made useful and effective revisions. As noted above the turn of the discussion in the manuscript is sensible and the language is clear. It is well written. On the question of entropy and tunnelling raised by this reviewer 3, the revisions do not fully satisfy the questions raised in my view. Entropy effects should be clarified as temperature dependent, and the effect on catalysis of tunnelling and entropy should be made clear. The response repeats the confusion between catalysis and reaction: k_{cat} measures the rate of reaction, it does not measure catalysis, which is the acceleration of the reaction rate. The need for clear distinction between reactivity and catalysis is discussed above and has been discussed by Warshel et al. The manuscript should be modified to avoid such confusion

Response/changes: First of all, Reviewer 4 may not have been aware of the fact that the original Reviewer 3 was fine with the revisions made (see above). Concerning entropy and the temperature dependence, we have responded to Reviewer 4 as well as we have done so in the earlier response to Reviewer 3. The temperature dependence of entropy is emphasized but it is also clarified that this fact doesn't invalidate the discussion of entropy at a specified temperature. The use of terminology was updated in an effort to become fully

rigorous, also in response to the original comments of Reviewer 4. Only to note, Reviewer 3 was satisfied with the responses given.

Note on author list

Removal of the manuscript part dealing with use of the phenomenological model has resulted in a change of authors during revision. Zdenek Petrasek is no longer an author because his contribution was exclusively on this part. A duly signed form by all authors (current and previous) is uploaded.

REVIEWERS' COMMENTS

Reviewer #4 (Remarks to the Author):

I thank the authors for the significant effort and seriousness that they have shown in revising the manuscript and in responding to my points and the previous reviews. This is altogether a most interesting discussion. The revisions that the authors have made have significantly improved the manuscript, the experimental quality of which was already clear. The paper is now publishable in my view. I would not want to delay publication by minor quibbles. This work represents an important contribution to an interesting and controversial area; the discussion and presentation is measured and balanced and the points made are well evidenced in the revised manuscript. This is an important contribution and the results will be of wide interest.

REVIEWERS' COMMENTS

Reviewer #4 (Remarks to the Author):

I thank the authors for the significant effort and seriousness that they have shown in revising the manuscript and in responding to my points and the previous reviews. This is altogether a most interesting discussion. The revisions that the authors have made have significantly improved the manuscript, the experimental quality of which was already clear. The paper is now publishable in my view. I would not want to delay publication by minor quibbles. This work represents an important contribution to an interesting and controversial area; the discussion and presentation is measured and balanced and the points made are well evidenced in the revised manuscript. This is an important contribution and the results will be of wide interest.

Response. We are grateful for the positive response to our revised manuscript. No changes were necessary in response to the comment of the Reviewer.